# Clonal reversal of ageing-associated stem cell lineage bias via a pluripotent intermediate

Martin Wahlestedt[1], Eva Erlandsson[1], Trine Kristiansen[1], Rong Lu[2], Cord Brakebusch[3,4], Irving L. Weissman[5], Joan Yuan[1,6], Javier Martin-Gonzalez[4] & David Bryder[1,6]

Ageing associates with significant alterations in somatic/adult stem cells and therapies to counteract these might have profound benefits for health. In the blood, haematopoietic stem cell (HSC) ageing is linked to several functional shortcomings. However, besides the recent realization that individual HSCs might be preset differentially already from young age, HSCs might also age asynchronously. Evaluating the prospects for HSC rejuvenation therefore ultimately requires approaching those HSCs that are functionally affected by age. Here we combine genetic barcoding of aged murine HSCs with the generation of induced pluripotent stem (iPS) cells. This allows us to specifically focus on aged HSCs presenting with a pronounced lineage skewing, a hallmark of HSC ageing. Functional and molecular evaluations reveal haematopoiesis from these iPS clones to be indistinguishable from that associating with young mice. Our data thereby provide direct support to the notion that several key functional attributes of HSC ageing can be reversed.

[1] Lund University, Medical Faculty, Institution for Laboratory Medicine, Division of Molecular Hematology, Klinikgatan 26, BMC B12, 221 84 Lund, Sweden. [2] Department of Stem Cell Biology and Regenerative Medicine, Eli and Edythe Broad Center for Regenerative Medicine and Stem Cell Research, Keck School of Medicine, University of Southern California, Los Angeles, California 90033, USA. [3] Biotech Research and Innovation Centre, Biomedical Institute, University of Copenhagen, 2200 Copenhagen, Denmark. [4] Core Facility for Transgenic Mice, Department of Biomedical Sciences University of Copenhagen, The Panum Institute, 6.4, Blegdamsvej 3B, 2200 Copenhagen, Denmark. [5] Institute of Stem Cell Biology and Regenerative Medicine, Stanford University School of Medicine, Stanford University, Palo Alto, California 94305, USA. [6] StemTherapy, Lund University. Correspondence and requests for materials should be addressed to D.B. (email: David.Bryder@med.lu.se).

Ageing associates with a profound predisposition for an array of diseases, which in the blood includes a higher prevalence for anaemia, leukaemia and compromised immunity[1]. While age-related diseases evidently can arise due to changes that compromise or alter the function of mature effector cells, this is harder to reconcile with organs such as the blood, that rely on inherently short-lived effector cells in need of continuous replenishment[1–3]. Rather, accumulating data have suggested that the *de novo* production of subclasses of haematopoietic cells shifts in an age-dependent manner[4–7], akin to that seen during more narrow time windows in early development[8]. These findings have to a large extent also challenged the classically defining criteria of haematopoietic stem cells (HSCs) as a homogenous population of cells with differentiation capacity for all haematopoietic lineages. Rather, the differentiation capacity of HSCs might be more appropriately defined by a continuous multilineage haematopoietic output, but which might not necessarily include the production of all types of blood cells at all points in time.

While many of the changes in the ageing adult are underwritten by alterations in HSC function[1], the individual constituents of the HSC pool can display a significant variation in function[4,9,10]. Apart from individual HSCs being preset differentially[5,6,11], which could gradually alter the composition of the HSC pool with age[5,6], other mechanisms leading to segmental changes within the HSC pool, including environmental influences, uneven proliferative rates and acquisition of DNA mutations in individual cells, are also possible[1–3]. Hence, by merely evaluating chronologically aged cell populations, the heterogeneity of individual cells is not accounted for.

The mechanisms that drive ageing at both the organismal and cellular level have attracted significant attention as they represent prime targets for intervention. For instance, prolonged health- and lifespan has been reported in a variety of model organisms by caloric restriction and/or by manipulating the IGF1 and mTOR axes[3]. Moreover, an increased function of aged cells by 'young'-associated systemic factors has been proposed[12]. Whether such approaches indeed reflect rejuvenation at a cellular level or

rather stimulate cells less affected by age is mostly unclear. This concern applies also to previous studies approaching the prospects of reversing cellular ageing by somatic cell reprogramming[13–15], which have typically failed to distinguish between functionally versus merely chronologically aged cells. To do this, there is a need to reliably define the function of the specific parental donor cell used for reprogramming, which necessitates evaluations at a clonal/single-cell level.

Here we approach these issues by genetic barcoding of young and aged HSCs that allows for evaluations, at a clonal level, of their regenerative capacities following transplantation. This allows us to establish that ageing associates with a decrease of HSC clones with lymphoid potential and an increase of clones with myeloid potential. We generate induced pluripotent stem (iPS) lines from functionally defined aged HSC clones, which we next evaluate from the perspective of their blood-forming capacity following re-differentiation into HSCs by blastocyst/morula complementation. Our experiments reveal that all tested iPS clones, including such that were originally completely devoid of T- and/or B-cell potential, perform similar to young HSCs both in steady-state (1° chimeras) and when forced to regenerate lymphomyeloid haematopoiesis in secondary transplantations. This regain in function coincides with transcriptional features shared with young rather than aged HSCs. Thereby, we provide direct support to the notion that several functional aspects of HSC ageing can be reversed to a young-like state.

## Results

### The clonal composition of the HSC pool as a consequence of age.

We first determined the clonal compositions of the HSC pools in young and aged mice by genetic barcoding of HSCs[9], followed by competitive transplantation (1° transplant) and retrospective tracking of their progeny long-term after transplantation (Fig. 1). In agreement with previous studies[7,10,16], peripheral blood (PB) analysis of these recipients revealed a distinct lineage distribution from aged HSCs, where the most striking features included

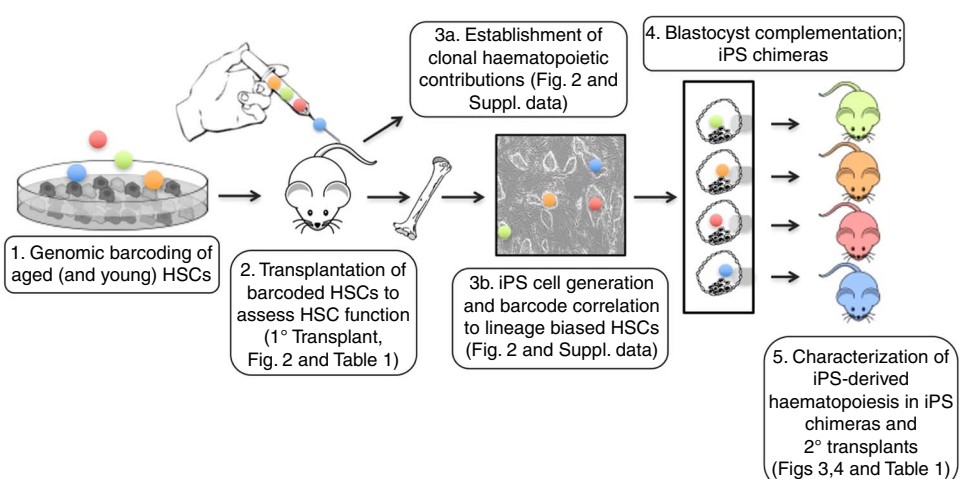

**Figure 1 | Experimental outline of the work.** In steps 1 and 2, young and aged HSCs were isolated, barcoded and competitively transplanted into lethally irradiated young hosts. Following the establishment and assessment of long-term haematopoiesis contributed from the barcoded cells (step 3a), peripheral B, T and myeloid cells, as well as erythroid progenitors from the BM, were isolated and subjected to deep sequencing to unravel their underlying barcodes (step 3b). Simultaneously, immature BM cells were isolated from a mouse transplanted with aged barcoded HSCs (1° transplant) completely lacking T-cell reconstitution (reduced T-cell generation is a hallmark of HSC ageing). The barcodes in the obtained iPS lines were investigated for overlap with the lineage potential of individual HSCs obtained in step 3a. Next, five iPS lines whose barcodes associated with a myeloid-lineage skewing were used to generate iPS chimeras by blastocyst and morula complementation (step 4). HSCs in iPS chimeras were investigated for both functional and molecular blood-forming parameters and their performance was compared to young (functional and molecular assays), middle-aged (molecular assays) and aged HSCs (molecular assays; step 5).

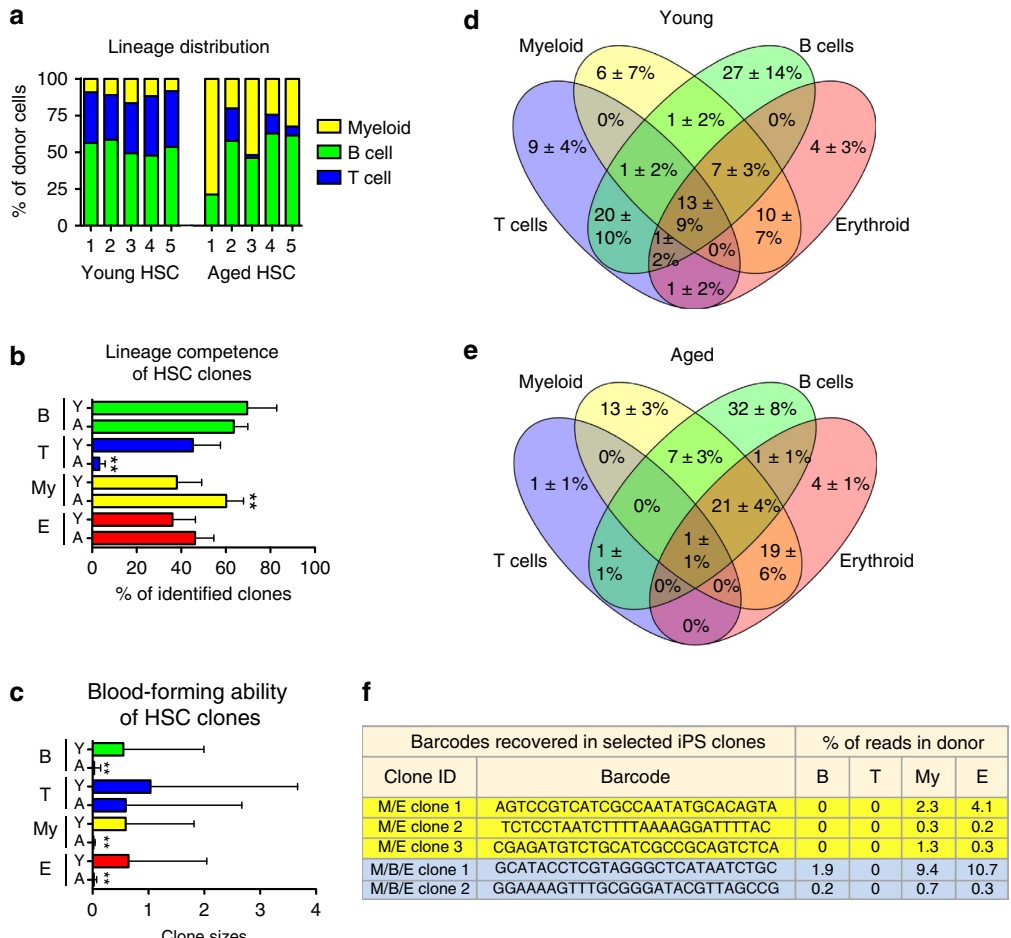

**Figure 2 | Clonal HSC dynamics as a consequence of age.** (**a**) Lineage composition of young and aged HSCs in the peripheral blood (PB) of individually repopulated mice 4 months after transplantation, with each bar depicting individual recipients (1° transplant). (**b**) Barcode distributions were established from FACS sorted donor derived B, T, myeloid and bone marrow (BM) erythroid progenitor cells from all transplanted mice 20–24 weeks after transplantation. Depicted is the barcode representation in each investigated blood cell lineage derived from young and aged HSCs. (**c**) Clone sizes of barcodes detected in **b**. Barcodes identified in the different lineages were intersected to determine the frequencies of each type of mono-, bi- and multipotent HSC clones. Venn diagrams show young (**d**) and aged (**e**) clone types, with the average frequency (%) ± s.d. of each clone type. (**f**) Donor-derived myeloid progenitors isolated from aged mouse 1 were reprogrammed into iPS cells, and iPS clones were subsequently analysed for barcode integration. Shown are barcode sequences and their frequencies (%) in the parental donor mouse for the five iPS lines selected for further experiments. The data are from one experiment. Error bars indicate mean ± s.d. ** indicates significant differences ( = P < 0.05, unpaired two-tailed t-tests in **b**,**c**). In **b**, the young and aged T-cell comparison yielded a P value of < 0.0001, and the young and aged myeloid cell comparison yielded a P value of 0.0064. In **c**, the comparisons of individual young and aged HSCs ability to generate B, myeloid and erythroid all yielded P vales < 0.0001.

a very low abundance of T cells and an increased frequency of myeloid cells (Fig. 2a). To investigate the clonal repopulation dynamics in detail, we next analysed the barcodes retrieved from peripheral B-, T- and granulocyte/myeloid cells, as well as from bone marrow (BM) erythroid progenitors (Fig. 2b–e; Supplementary Fig. 1). In agreement with a previous barcoding study of aged HSCs[4], both young and aged HSC clones contributed actively to haematopoiesis (Supplementary Fig. 1a). Of these clones, the proportion that contributed to B cells and erythropoiesis was similar regardless of age. A higher frequency of aged clones contributed to myelopoiesis (38% of young, 60% of aged) while, strikingly, very few aged HSC clones contributed to the T-cell lineage (45% of young, 3% of old; Fig. 2b). When investigating the magnitudes of blood cell output, aged HSCs generated markedly fewer B cells, erythroid cells and myeloid cells compared to their young counterparts (20-, 29- and 39-fold smaller average clone sizes, respectively; Fig. 2c; Supplementary Fig. 1b). Of note, the few T-cell competent HSCs that persisted

with age produced a T-cell output similar in magnitude to that from young HSCs (Fig. 2c).

As a second part of our analysis, we intersected the barcode data from all lineages to infer the clonal combinatorial contributions from young and aged HSCs (Fig. 2d–e; Supplementary Fig. 1b,c). While HSCs contributed to a spectrum of different lineage outputs regardless of age, myeloid-restricted clones and bipotent clones with combined myeloid and erythroid potential increased in frequency with age (6 ± 7% (mean ± s.d.) versus 13 ± 3% and 10 ± 7% versus 19 ± 6% for young and aged HSCs respectively), while all combinations that involved T cells were reduced (Fig. 2d,e). In addition, our analyses revealed that HSC ageing associated with a pronounced decrease in the ability of candidate HSCs to contribute to all of the four lineages evaluated (13 ± 9% (mean ± s.d.) of young and 1 ± 1% of aged clones). However, when investigating the output of these multipotent clones, they were responsible for vast majority of the T-cell output observed in aged mice (Supplementary Fig. 1c).

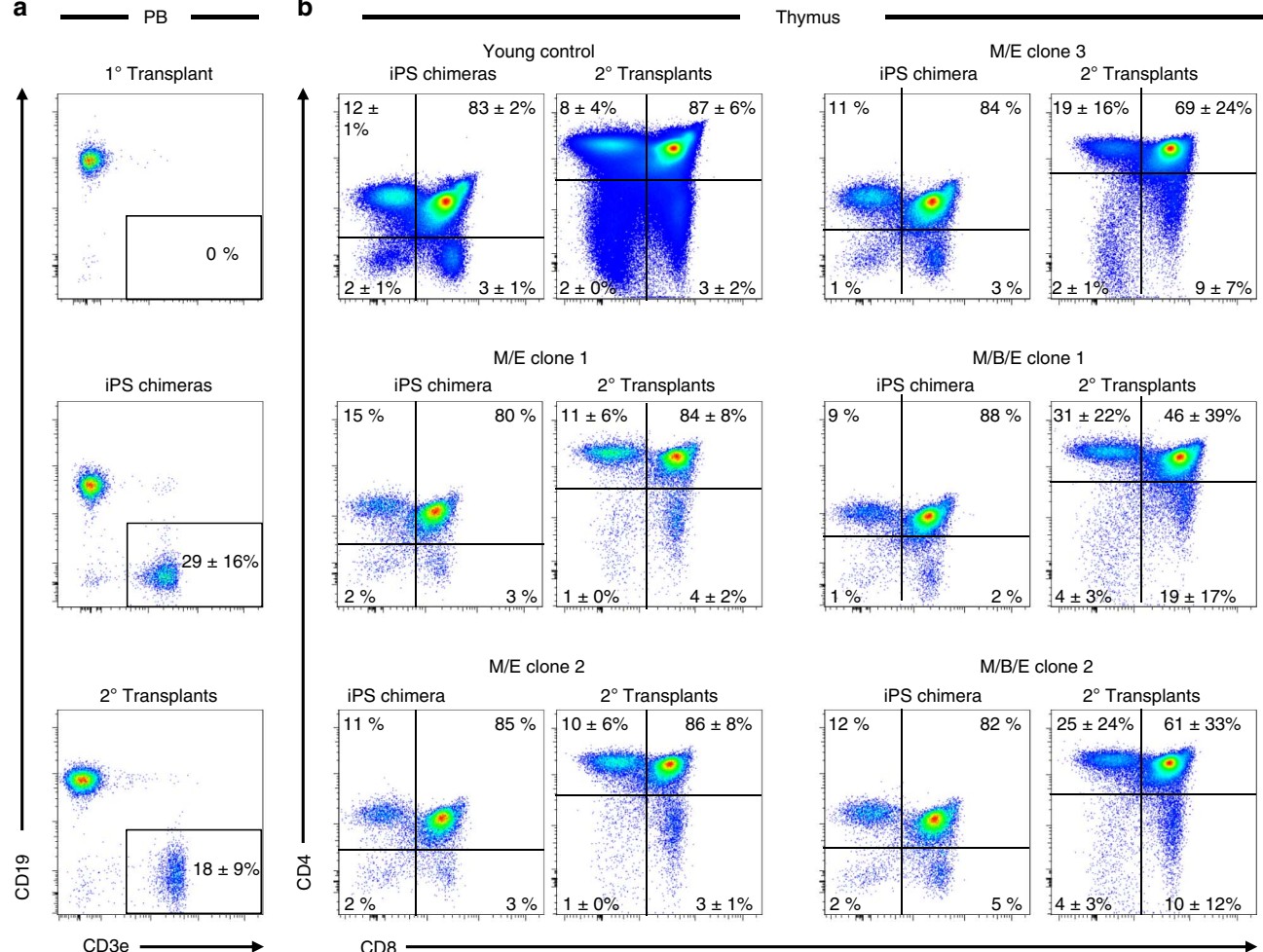

**Figure 3 | Regain of the age-associated loss of T-cell potential.** (**a**) Peripheral T-cell generation from barcoded HSCs in 1° transplants, iPS chimeras and in 2° transplants. The average frequency (%) ± s.d. among all test cells is depicted ($n = 5$ and 22 mice for the iPS and 2° transplants, respectively). The cells in the FACS plots are pregated on single, viable, CD45.2$^+$, GFP$^+$, CD11b$^-$ cells and on single, viable, CD45.2$^+$, CD11b$^-$ cells for blastocyst (iPS chimeras) and 2° transplants. (**b**) T-cell generation in the thymi of iPS chimeras and in 2° transplants. The average frequency (%) ± s.d. of CD4$^-$CD8$^-$, CD4$^+$CD8$^-$, CD4$^+$CD8$^+$ and CD4$^-$CD8$^+$ among all test cell-derived CD3$^+$ thymocytes is denoted in the corresponding gates. The cells are pregated on single, viable, lineage$^-$, CD3e$^+$, CD45.1$^+$/CD45.2$^+$ for young control and CD45.2$^+$ iPS-derived cells. Data are from one experiment (primary and secondary chimeras) for each iPS line.

Altogether, these data demonstrated that the altered output of mature effector cells with age is underwritten by distinct shifts in the abundance of different HSC clone types, but also in the magnitude, whereby individual HSC clones contribute to haematopoiesis. At the same time, these data established that ageing does not affect all HSCs in a similar manner.

**Functional reversal of aged HSC clones.** We next challenged the heritability of the ageing phenotype by generating iPS cells from a mix of aged barcoded progenitors from one of the original recipients that completely lacked donor-derived barcoded T cells (Figs 2a and 3a, aged HSC mouse #1). These parental cells represent the immediate progeny of the barcoded HSCs (Fig. 1) and were chosen over HSCs, as we have been unable to generate iPS cells from highly purified HSCs[13], likely owing to the high level of dormancy of stringently purified HSCs[17]. This approach led to the generation of 20 barcoded iPS lines. Following retrospective analysis of the retrieved barcodes to their haematopoietic lineage affiliations, we identified three unique barcodes originating from HSCs with a combined myeloid and erythroid, but no B- or T-cell potential (M/E clones 1–3) and two

additional myeloid-biased barcodes with myeloerythroid, B- but no T-cell potential (M/B/E clones 1–2; Fig. 2f and Methods). Next, these five barcode-assigned iPS lines were injected into CD45.1$^+$CD45.2$^+$ blastocysts or morulas, followed by implantation into pseudopregnant mothers (Fig. 1). Thereby, we could directly investigate the iPS-derived (CD45.2$^+$ CD45.1$^-$) haematopoiesis in primary chimeric mice (iPS chimeras) with simultaneous comparison to that of the endogenous (blastocyst/morula; young; CD45.1$^+$CD45.2$^+$)-derived HSCs. We chose this strategy rather than comparing to iPS-derived cells from young HSCs as prior work had established that the reprogramming *per se* does not seem to alter the lineage competence of young HSCs[13]. This revealed that all iPS clones evaluated generated not only myeloid cells but also B and T cells (Table 1).

To assess the haematopoietic repopulation potential of the iPS-derived cells, we next transplanted BM cells from the blastocyst/morula chimeras into lethally irradiated hosts (Fig. 1, 2° transplant). Because of the mixture of iPS-derived and endogenous in primary chimeras, these experiments are by nature competitive. This showed that both the quantitative

**Table 1 | Lineage distribution of barcoded clones before and after reprogramming.**

| Aged parental clone | Origin | Overall haematopoietic chimerism (% ± s.d.) | Overall relative chimerism (% ± s.d.) | Lineage distribution of test cells in 1° transplants, iPS chimeras and 2° transplants (% ± s.d.) | | | Lineage distribution of young cells (blastocyst/morula-derived; % ± s.d.) | | |
|---|---|---|---|---|---|---|---|---|---|
| | | | | T cell | B cell | Myeloid | T cell | B cell | Myeloid |
| M/E clone 1 (myeloid only) | 1° Transplant | NA | NA | **0** | **0** | 100 | NA | NA | NA |
| | iPS chimera 2° Transplant, iPS origin ($n=4$) | 26.9 25.8 ± 3.6 | 100 113.9 ± 45.4 | 11.2 8.7 ± 4.9 | 60.6 75.4 ± 10.9 | 28.2 15.9 ± 8.9 | 45.5 22.7 ± 9.5 | 32.3 62 ± 6.2 | 22.2 15.3 ± 3.5 |
| M/E clone 2 (myeloid only) | 1° Transplant | NA | NA | **0** | **0** | 100 | NA | NA | NA |
| | iPS chimera 2° Transplant, iPS origin ($n=4$) | 25.5 18.6 ± 3.3 | 100 73.1 ± 12.8 | 20.1 10.1 ± 1.7 | 60.9 76.9 ± 6.1 | 19 13.1 ± 6.8 | 58.5 26 ± 4.4 | 26.1 61.5 ± 3.6 | 15.3 12.5 ± 4.2 |
| M/E clone 3 (myeloid only) | 1° Transplant | NA | NA | **0** | **0** | 100 | NA | NA | NA |
| | iPS chimera 2° Transplant, iPS origin ($n=4$) | 55.4 60.9 ± 10.4 | 100 109.9 ± 18.7 | 32.9 22.3 ± 6 | 36.8 60.9 ± 1.1 | 30.3 16.8 ± 6.4 | 55.8 42.7 ± 5.2 | 27.2 41 ± 7.2 | 17 16.3 ± 6.3 |
| M/B/E clone 1 (T-cell deficient) | 1° Transplant | NA | NA | **0** | 16.5 | 83.5 | NA | NA | NA |
| | iPS chimera 2° Transplant, iPS origin ($n=5$) | 33.7 27.6 ± 9.7 | 100 79.7 ± 32.8 | 11.5 22.4 ± 6.2 | 69.2 64.7 ± 10.1 | 19.3 13 ± 5.9 | 36.9 43.7 ± 6.1 | 51.8 40.7 ± 7.8 | 11.3 15.6 ± 2.3 |
| M/B/E clone 2 (T-cell deficient) | 1° Transplant | NA | NA | **0** | 24.5 | 75.5 | NA | NA | NA |
| | iPS chimera 2° Transplant, iPS origin ($n=5$) | 22.4 21.4 ± 4.1 | 100 71.9 ± 31.5 | 29.5 12.8 ± 3.7 | 31.7 78.8 ± 3.3 | 38.8 8.3 ± 2.6 | 62.3 29.8 ± 5.3 | 22.3 54.3 ± 1.7 | 15.4 15.9 ± 6.3 |

BM, bone marrow; FACS, fluorescence-activated cell sorting; GFP, green florescent protein; iPS, induced pluripotent stem; NA, not applicable; PB, peripheral blood.
The PB lineage distribution of the parental HSCs used for somatic cell reprogramming was determined by the barcode read distribution in the different lineages obtained from barcode sequencing of the 1° transplanted mouse, and in iPS chimeras and in 2° transplants by PB FACS analysis (frequencies determined among CD45.2$^+$CD45.1$^-$ cells). GFP was not used in analysis of iPS-derived haematopoiesis due to a large degree of viral silencing coinciding with the iPS-cell generation. The relative overall PB iPS chimerism maintained following transfer of iPS BM in 2° transplants is also provided. Shown values are expressed as average values ± s.d. The data are from one experiment for each iPS line for iPS chimeras and 2° transplants. Bold entries denote the complete absence of detectable donor-derived B/T or T cells from evaluated clones in primary transplants.

(levels of donor-derived cells) and qualitative (contribution to each assessed lineage) properties from each of the iPS-derived HSCs (iPS-HSCs) were similar to that of the endogenous (young) control HSCs (Table 1). Given the complete lack of T-cell lineage contribution of the parental HSCs (Figs 2f and 3a), we complemented our investigations by more detailed analysis of T-cell development in the thymi of both iPS and 2° transplants (Fig. 3). This revealed comparable frequencies of CD3$^+$CD4$^-$CD8$^-$, CD3$^+$CD4$^+$CD8$^+$, CD3$^+$CD4$^+$CD8$^-$ and CD3$^+$CD4$^-$CD8$^+$ cells to the endogenous (young) control cells (Fig. 3b) and verified persistent re-establishment of T-cell competence from T-cell incompetent parental donor cells.

Having established that the haematopoiesis derived from iPS cells was functionally equivalent to young haematopoiesis in terms of differentiation and regenerative potential following transplantation, iPS-HSCs were next interrogated more directly. Apart from alterations in the lineage output of HSCs, ageing coincides with a numerical expansion of the HSC pool[7,10,16]. We therefore first investigated the frequency of HSCs in iPS chimeras (Fig. 4a). This revealed that the iPS-HSC frequency was equivalent to that of the young controls (Fig. 4b). In addition, the iPS-HSC pool was not, as opposed to the aged HSCs originally barcoded, dominated by cells that express high levels of CD150 and instead expressed levels of this marker comparable to young controls (Supplementary Fig. 2a,b). We thereafter applied a multiplexed single-cell quantitative PCR with reverse transcription approach on sorted iPS-HSCs for a panel of

genes that are differentially expressed with HSC ageing (Fig. 4c; Supplementary Table 1). Principal component analyses were used to evaluate their relationships to normal young (endogenous control cells from each iPS chimera) or aged HSCs. We here also included HSCs isolated from normal mid-aged (11 months old) mice, reasoning that this potentially could unveil a phenotype intermediate to that of young and aged HSCs. This revealed two distinct patterns; most middle-aged and aged HSCs clustered together and away from iPS-HSCs and the young HSCs, the latter two cell types clustering tightly together in all instances (Fig. 4d). Finally, although rather few cells were analysed for mid-aged and aged HSCs, more mid-aged HSCs overlapped with young and iPS-HSCs, suggesting that the composition of the HSC compartment changes gradually with age. Collectively, these data indicate that rejuvenation appears complete with the measured signatures of chronological age being erased.

## Discussion

Most organs contain adult/somatic stem cell populations that function to maintain homeostasis throughout life. Given the growing evidence that adult stem cells are not spared from ageing, it appears reasonable that the widespread decline in tissue function with age is influenced by an altered function of their associated stem cells. At the same time, any pool of ageing stem cells needs be regarded as potentially heterogeneous; either because individual stem cells might be preset differentially from young age, the different experiences met by individual stem cells

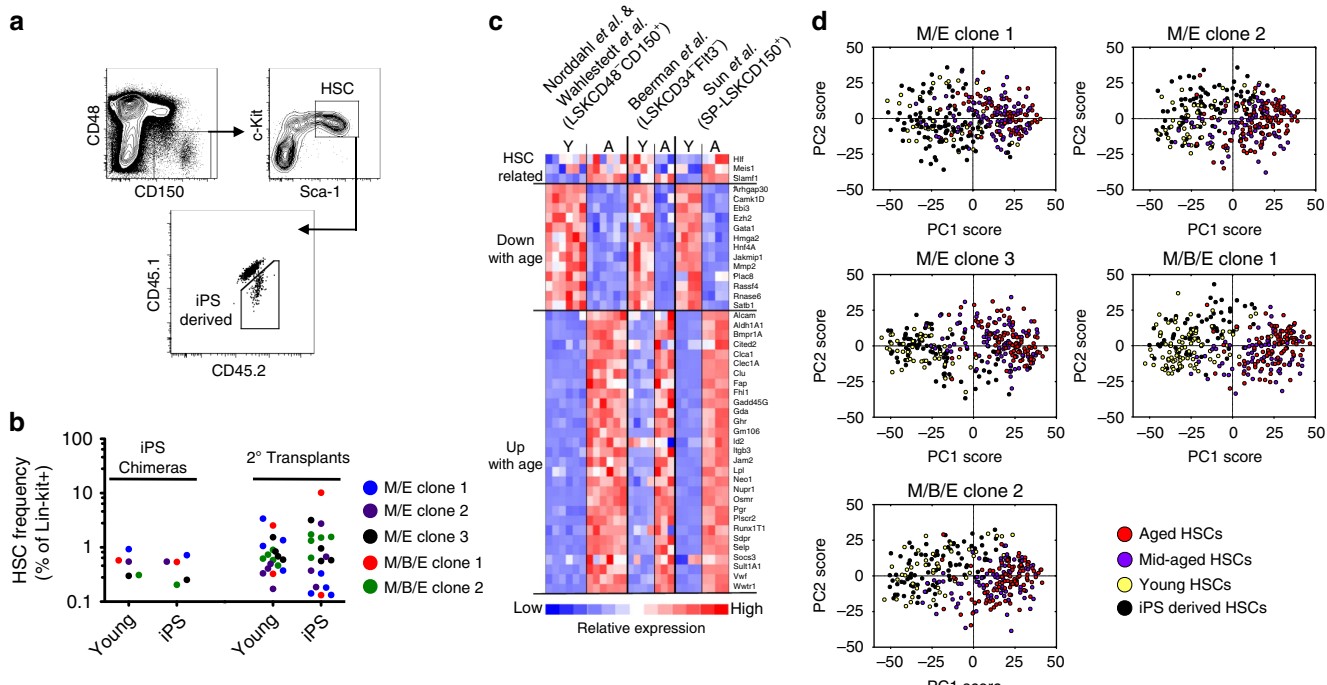

**Figure 4 | IPS-derived HSCs are appropriately maintained and display a gene expression pattern reminiscent of young HSCs.** (**a**) FACS gating strategy to distinguish iPS-HSCs from young endogenous HSCs in iPS chimeras (pregated on single, viable and lineage-negative cells.). (**b**) The frequencies of HSCs were determined among iPS and young Lin$^-$c-Kit$^+$ bone marrow cells in iPS ($n = 1$ chimera per iPS clone) and 2° transplants ($n = 4, 4, 4, 2$ and 5 mice for M/E clone 1, M/E clone 2, M/E clone 3, M/B/E clone 1 and M/B/E clone 2, respectively). (**c**) Heatmap showing the relative expression of the 45 genes selected for single-cell PCR with reverse transcription (RT–PCR) in previously published data sets of young and aged HSCs. (**d**) Single iPS-HSCs and chronologically young endogenous HSCs were isolated from each iPS chimera and subjected to multiplexed quantitative RT–PCR. Principal component analyses for each analysed iPS-derived HSCs and young control HSCs isolated from the same iPS chimera ($n = 90, 90, 89, 67, 88$ cells and $47, 47, 47, 91, 45$ cells for the iPS-HSC and young HSCs isolated from M/E clone 1, M/E clone 2, M/E clone 3, M/B/E clone 1 and M/B/E clone 2, respectively) as compared to middle-aged ($n = 92$ cells) and aged HSCs ($n = 90$ cells). Data are from one experiment for each iPS line and endogenous blastocyst control cells, and from one experiment for middle-aged and aged HSCs.

throughput ontogeny, or a combination of these facts. Hence, it is pivotal to be able to approach functionally as opposed to merely chronologically aged cells when investigating the prospects for reversal of cellular ageing.

Recently, correlations between human haematopoietic ageing and clonal expansions of haematopoietic cells carrying mutations in tumour suppressor genes such as *DNMT3A* and *TET2* have been observed[18–20]. However, although such events of mutation-driven clonal haematopoiesis likely can contribute to the establishment of a preleukemic environment[21], their contributions to the shortcomings of human haematopoietic ageing in general is less clear. As somatic cell reprogramming, as here, mediates an epigenetic 'reset' of the parental cells, but should leave any underlying DNA mutations unrepaired, our results fail to support the view that accumulations of DNA mutations in genes critical for haematopoiesis would be the principal/only mechanism for aged-dependent functional decline of HSCs. Instead, our results favour an altered epigenome to be a major contributor to HSC ageing. This does not exclude that mechanisms involving macromolecular damage/insult might contribute to other aspects of HSC ageing not approached here. For instance, the nature of our experiments does not allow us to assess age-induced changes that lead to a direct cellular loss. At the same time, the aged epigenome can still be strongly rooted, which is not least supported by the recent data, suggesting that direct conversion of aged fibroblasts into neurons rendered cells with a retained aged transcriptome[15]. Therefore, a rather complete epigenomic 'reset', like that shown here, might be

needed to normalize the alterations that arise and govern cellular ageing. While it remains to be determined how this could be achieved in a more tractable manner, we believe our results should be encouraging for the prospects of such efforts, which in the long term would pave the way for the development of therapeutic modalities aimed at achieving overall healthier late stages of life.

## Methods

**Mice.** *Rosa26*$^{rtTA}$;*Col1a1*$^{4F2A}$ mice (4F2A mice) were kindly provided by Dr Rudolf Jaenisch and acquired via the Jackson laboratory (stock number 011004). Cells from these mice express *Oct4*, *Klf4*, *Myc* and *Sox2* upon doxycycline supplementation[22]. All mice were on a C57BL/6 background and carried the congenic marker combinations CD45.1, CD45.2 or F1CD45.1 × CD45.2. The animals were housed at animal facilities at Lund University and Copenhagen University. All animal experiments were performed with consent from local ethical committees. Both male and female mice were used for experiments.

**Immunophenotypic analyses and cell sorting.** For analysis and isolation of HSCs and myeloerythroid progenitors, single-cell suspensions of BM cells were lineage-depleted using biotinylated antibodies (anti-B220, -CD4, -CD8, -CD11b, -Gr-1 and -TER-119) and anti-biotin magnetic beads (according to the manufacturer's instructions, Miltenyi Biotec) before incubation with fluorescently labelled antibodies and streptavidin. Alternatively, BM cell suspensions were first c-kit-enriched using magnetic beads (Miltenyi Biotec), followed by staining with fluorescently labelled streptavidin and antibodies directed against the epitopes specific for the cells of interest. HSCs were identified as Lin$^-$ Sca-1$^+$ c-Kit$^+$ CD48$^-$ CD150$^+$ and colony forming unit erythroid cells (CFU-Es) as Lin$^-$ Sca-1$^-$ c-Kit$^+$ CD41$^-$ CD105$^+$ CD150$^-$ (ref. 23). To investigate thymic T-cell subsets, cell suspensions were depleted of B220-, CD11b-, Gr-1- and TER-119-positive cells using biotinylated antibodies and anti-biotin magnetic beads (Miltenyi Biotec). Next, the

cells were stained with fluorescently labelled antibodies directed against CD3, CD4 and CD8. For blood analysis and sorting of mature effector cells, PB was collected from the tail vein and red blood cells were sedimented with 1% Dextran T-500 (Sigma). Remaining red blood cells were lysed with ammonium chloride before staining with antibodies (CD3, CD11b and CD19). When preparing the BM, thymus and PB samples, CD45.1 and CD45.2 antibodies were included to distinguish iPS-derived cells from endogenous blood cells, and propidium iodide (used according to the manufacturer's instructions, ThermoFisher Scientific) to exclude dead cells. Cells were sorted on a FACS AriaII or III cell sorter and analysed on an LSRII or an LSR Fortessa (Becton Dickinson) made available at the FACS core at Lund Stem Cell Center. Representative fluorescence-activated cell sorting (FACS) profiles and gating strategies are shown in Supplementary Fig. 3 and the antibodies used are listed in Supplementary Table 2.

**Investigations into the clonality of HSC ageing.** *1° Transplants.* Twenty-three-month-old 4F2A and 8–10-week-old C57BL/6 HSCs (CD45.2$^+$ or CD45.1$^+$, respectively) were isolated and transduced overnight with a lentiviral vector barcode library[9], resulting in 29% and 31% transduction, respectively (as determined by a GFP reporter gene). Sixteen hours later, 6,900 aged or 1,000 young cells were transplanted into lethally irradiated (950 cGy) 8–10-week-old CD45.1$^+$ or CD45.1$^+$/CD45.2$^+$ recipients (five mice per group) without prior selection for transduced cells, along with 300,000 unfractionated competitor cells (CD45.1$^+$ or CD45.1$^+$/CD45.2$^+$, respectively) to ensure survival and provide competitor cells. Aged HSCs were transplanted in higher numbers to ensure robust reconstitution, since it is well established that aged candidate HSCs display a markedly reduced regenerative potential compared to their young counterparts[1,4–6,10,16]. Reconstitution of the transplanted cells was monitored periodically in PB. Long term after transplantation (20–24 weeks), donor-derived CD19$^+$ B cells, CD3e$^+$ T cells, CD11b$^+$ myeloid cells from the PB and CFU-Es from the BM were isolated from each recipient and suspended in lysis buffer (100 mM Tris-Hcl pH 8.5, 1 mM EDTA pH 8.0, 200 mM NaCl, 0.2% SDS and 500 µg ml$^{-1}$ Proteinase K), incubated for 2 h at 56 °C, followed by 10 min at 95 °C. The genomic DNA (gDNA) in the lysates was next purified using Ampure XP beads (Beckman Coulter), and 5,000–25,000 cell equivalents of gDNA were used as templates in PCR reactions designed to amplify a 238 bp amplicon containing the barcode sequence, sample multiplex information and IonTorrent sequencing adaptors. In cases where an excess of 25,000 cells were obtained, PCR products were pooled following the PCRs. The reactions were amplified using Q5 High-Fidelity DNA Polymerase (NEB) with cycling conditions: 30 s at 98 °C, 32 cycles of 10 s at 98 °C and 60 s at 72 °C, followed by a final 2 min incubation at 72 °C. Following PCR, the samples were purified using Ampure XP beads, concentrations measured using a Qubit Fluorometer (Life Technologies) and DNA integrity measured with a Bioanalyzer (Agilent Technologies). Equimolar amounts of DNA from all young and aged cell types were independently pooled and sequencing was next performed at NGI Uppsala (Uppsala Genome Center, Scielifelab) on an Ion Torrent PGM instrument (Life Technologies) using the 400 bp chemistry with Ion 314 chips.

Following sequencing, all reads containing the complete barcode (27 bp) and library ID sequence (7 bp)[9] were extracted from the FASTQ files using custom R scripts, before being further filtered to exclude wrongly called independent barcodes due to sequencing errors, as described[24] (see Supplementary Table 3 for sequencing coverage and barcode filtration details). Next, the number of reads per barcode was expressed as percentage of the total barcode reads in that sample. Thereafter, the barcodes present in the different lineages were intersected to determine the representation of HSC clone types. The clone size, or the amount by which each HSC (corresponding to a unique barcode) contributed to individual lineages, was calculated by dividing its frequency among all reads in a given lineage (for each mouse) by the % of GFP$^+$ cells among the donor cells for the same lineage. A cutoff for detection was introduced according to the number of GFP$^+$ cells that were initially transplanted (estimated based on the number of cells transplanted and the transduction levels). If a barcode was found represented below 1 divided with the number of GFP$^+$ cells transplanted, it was classified as noise and omitted from all subsequent analyses. The list of barcodes along with their read counts in each mouse is provided in Supplementary Data 1.

**Derivation of iPS cells.** CD45.2$^+$Lin$^-$Sca-1$^-$c-Kit$^+$GFP$^+$, excluding CFU-Es, were isolated from aged HSC transplanted mouse 1 (Fig. 1) and maintained for 3 days in embryonic stem cell (ES) medium (DMEM (Invitrogen) containing 15% FCS, penicillin/streptomycin (Invitrogen), 1 mM sodium pyruvate (Invitrogen), 0.1 mM β-mercaptoethanol (Invitrogen), 1× MEM NEAA (Invitrogen) and 10$^3$ U ml$^{-1}$ LIF (Millipore, Billerica, MA)), supplemented with 50 ng ml$^{-1}$ mSCF, 10 ng ml$^{-1}$ hTPO, 5 ng ml$^{-1}$ mIL-3, 5 ng ml$^{-1}$ granulocyte colony-stimulating factor, 5 U ml$^{-1}$ EPO (all from PeproTech). Doxycycline (2 µg ml$^{-1}$; Sigma-Aldrich, St Louis, MO) was added to the culture after 24 h and the cells were transferred onto irradiated mouse embryonic fibroblasts after 72 h and maintained with doxycycline in ES medium (without haematopoietic cytokines). Emerging clones were manually picked under a light microscope between days 15 and 21, and were maintained in the absence of doxycycline during expansion. Candidate iPS clones were investigated for EPCAM and SSEA-1 expression[13], and iPS cells were FACS-sorted into lysis buffer for gDNA extraction. IPS clones containing

barcodes were subjected to Sanger sequencing (Eurofins) to determine the identity of the barcode and matched against the IonTorrent sequencing data to determine the HSC clone of origin for each iPS clone. From the 20 iPS lines generated and determined to carry a barcode, we identified nine unique barcodes, out of which five corresponded to myeloid-biased clones. The barcode of M/E clone 1 was found in six separate iPS lines, and the barcode of M/B/E clone 1 was found in three separate iPS lines. Other iPS lines generated from myeloid-biased HSCs were represented by unique iPS lines. Out of the four remaining barcodes, two barcodes could not be connected to any haematopoietic lineage affiliation (one of these barcodes was found in six separate iPS lines and the other in a unique line). Finally, two lines contained truncated barcodes (18 bp and 26 bp respectively). No further evaluations were made of these latter iPS lines.

**Generation and characterization of clone-specific chimeric mice.**
*iPS chimeras.* The iPS clones of choice (originally CD45.2$^+$) were injected into morulas and blastocysts from CD45.1$^+$/.2$^+$ mice at the Core Facility for Transgenic Mice (Copenhagen University, Denmark). Resulting chimeras were investigated for agouti fur colour, by PCR against the specific barcode and by the presence of CD45.2$^+$ single-positive cells in the PB. At 11–14 weeks of age, chimeric mice were killed, and their BMs, PBs and thymi were collected for FACS analysis and sorting.

*2° Transplants.* To investigate the competitive ability of iPS-derived HSCs, four to five lethally irradiated (950 cGy) 8–14-week-old CD45.1$^+$ or CD45.1$^+$/.2$^+$ F1 mice were transplanted with 2 × 10$^6$ unfractionated BM cells from each iPS chimera. Multilineage reconstitution was assessed periodically by FACS of PB.

To investigate the molecular resemblance of the iPS-derived HSCs to HSCs of different chronological ages, we first selected 3 highly expressed HSC-related genes, 3 reference genes (Actb, Hprt and Gapdh) and 42 genes differentially expressed in HSCs as a consequence of age. The latter category was obtained by analysis of previously published microarray data of young and aged steady-state HSCs (six arrays per age). Data were preprocessed by extracting probe level expression values using RMA[25] and further analysed using dChip[26] by filtering out probes with a lower expression than 50 in all subsets to eliminate noise and with a differential expression lower than 1.5-fold. Next, up- and downregulated genes were used as gene sets for a gene set enrichment analysis[27] of the steady-state young and aged HSC arrays and 42 genes were selected from the leading edge gene lists of these analyses.

The age-associated differential expression of the selected genes was confirmed in independently generated microarray and RNA sequencing (RNA-seq) expression sets. The microarray data set was preprocessed as described above and the analysis of the RNA-seq data set was performed using provided fragments per kilobase of exon per million fragments mapped (FPKM) values. The heatmap in Fig. 3c was generated in dChip. Next, iPS-HSCs and endogenous young HSCs isolated from the same iPS chimeras (11–14 weeks of age), middle-aged (11 months of age) and aged HSCs (22 months of age) were used for multiplex quantitative PCR with reverse transcription analyses using the Fluidigm Biomark platform as previously described[28]. In brief, single HSCs were sorted into 5 µl lysis buffer (10 mM TRIS-HCl pH 8.0, 0.1 mM EDTA, 0.1 U µl$^{-1}$ SUPERase-In (Clontech) and 0.5% NP40 (Igepal-CA630 Sigma Aldrich, St Louis, MO)). Sorted plates were stored at −80 °C until further processing. After cDNA synthesis using qScript cDNA SuperMix (Quanta Bioscience, Beverly, MA) each sample was pre-amplified in multiplex with target specific primers (Supplementary Table 1) using TATAA PreAmp GrandMaster Mix (TATAA Biocenter, Gothenburg, Sweden) for 22 cycles of 96 °C for 15 s and 60 °C for 6 min. After amplification, the PCR products were treated with Exonuclease I (NEB, Ipswich, MA) to remove unused primers. Gene expression analysis was next performed using the 48.48 Dynamic array Integrated Fluidic Circuits (48 IFC) on the Biomark HD platform (Fluidigm, San Francisco, CA). For each gene assay, 2.5 µl 2× Assay loading reagent (Fluidigm, San Francisco, CA) was mixed with 2.5 µl of diluted assays from Fluidigm and in-house designed assays (forward and reverse primers mixed at a final concentration of 5 µM). Samples were diluted 5× in low EDTA TE buffer (10 mM TRIS-HCl pH 8.0 and 0.1 mM EDTA) before loading the 48 IFC. A premix of 2.7 µl diluted sample and 3.3 µl of 2× SsoFast EvaGreen Supermix with low ROX (Biorad, Hercules, CA) and 20× DNA-binding Dye (Fluidigm, San Francisco, CA) was prepared for each sample. Five microlitre of each sample and assay were loaded into individual sample and assay inlets on the 48 IFC. Using the IFC controller MX (Fluidigm), the samples and assays were loaded into the reaction chambers and the 48 IFC was then transferred to the BioMark HD unit for quantitative PCR. The Fast PCR program included an initial hot start step of 1 min at 95 °C and then 30 cycles of 96 °C for 5 s and 60 °C for 20 s. Upon completion of the PCR, a melting curve analysis was performed with a ramp from 60 to 95 °C at 1 °C/3 s. Data were analysed using the Fluidigm Real-time PCR Analysis software using the Linear (Derative) Baseline Correction method and the Auto (Global) Ct Threshold Method. The Ct values determined were exported to SCExV[29] and Excel for further processing.

**Statistics.** Data were analysed using Microsoft Excel (Microsoft, http://www.microsoft.com) and Prism (GraphPad Software). All FACS data were analysed using the Flowjo software (TreeStar, Ashland, OR, http://www.flowjo.com). Venn diagrams were generated using Venny[30]. Significance values were calculated

by Student's two-tailed *t*-test and a *P* value of $< 0.05$, indicated as **, was used to determine significance. No statistical method was used to predetermine sample size and experiments were not randomized. No data were excluded from analysis. The investigators were not blinded to allocation during experiments or outcome assessment.

**Data availability**. The microarray data sets used to determine genes deregulated with age are found in Gene Expression Omnibus (GEO, https://www.ncbi.nlm.-nih.gov/geo) under accession numbers GSE27686 and GSE44923. The microarray and RNA-seq data sets used to independently validate the ageing gene lists can also be found in GEO under accession numbers GSE55525, GSE43729 and GSE47817.

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

## Acknowledgements

This work was supported by project grants to D.B. from the Swedish Cancer Society, the Swedish Medical Research Council, the Swedish Pediatric Leukemia Foundation, Knut and Alice Wallenberg foundation, and an ERC consolidator grant (615068). We would like to acknowledge Gerd Sten for expert technical assistance and Shamit Soneji, Stefan Lang and Sofia Adolfsson for suggestions on bioinformatics analyses.

## Author contributions

M.W. and D.B. designed the study. M.W. and E.E. performed experiments and J.M.G. and C.B. executed and supervised blastocyst/morula injections. T.K., R.L., I.L.W. and J.Y. were involved in the design and interpretation of the barcoding experiments and provided critical reagents. D.B. conceived and supervised the study, and wrote the paper together with M.W.

## Additional information

**Competing financial interests:** The authors declare no competing financial interests.

**Publisher's note**: 

