## [Peer Review File · Nature Communications]

Reviewers' comments:

Reviewer #1 (Remarks to the Author):

In this elegant study performed by the Bryder group, the authors have determined on a clonal level that function potential of aged primitive hematopoietic cells can be reset through reprogramming. Using bar-coding techniques to identify lineage output as well as overall contribution, the authors were able to demonstrate that multi-lineage potential, can be restored to cells derived from aged, lineage biased HSCs. The implications of this study are profound as it establishes that cellular reprogramming — which is orchestrated at the level of the epigenome — can reset the epigenome of aged murine HSCs to restore functional potential otherwise lost through aging. In so doing, the study firmly establishes that epigenetic remodeling is a central driver of HSC aging that is amenable to reprogramming.

Comments:

The restoration of overall reconstitution potential is not well characterized in this study. To demonstrate improved reconstitution potential, the authors should perform competitive transplants with the iPS derived HSCs as aged HSCs- with no competition can fully reconstitute an irradiated recipient mouse. The authors should define if transplanted BM from the blastocyst/morula was done in a competitive setting? If so, they should provide the total reconstitution frequency in table 1 for comparison to young transplanted. Also, the data for the control young transplants should be included in table 1.

The authors have previously published on lineage-biased HSCs, defining these populations using CD150/ SLAMF1 expression defining these subsets. The aged population of myeloid biased subsets express high levels of CD150. Were the aged M/E clones expressing high levels of CD150, and during the reprogramming did the expression levels of CD150 change along with lineage contribution?

Could the authors clarify the number of barcoded-HSCs transplanted /or starting number of HSCs transduced if the GFP+ cells were not purified before transplant. This point is relevant as the authors state similar numbers of HSCs contribute to overall hematopoiesis in both the young and aged mice.

Perhaps a mistake in the methods, but the manuscript states that aged-HSCs were reprogrammed to iPS (line 135) yet in the methods, myeloid progenitors (LSK cKit+Sca1-) are used for the generation of iPS lines. While myeloid progenitors are of course derived from stem cells, the authors should nonetheless clarify. Similarly, in the generation of iPS lines, were there many unique barcodes that were found in the progenitor population that were not read out in the peripheral blood?- i.e were there HSC clones that were active in contributing to primitive compartments, but did not contribute to peripheral blood / erythroid progenitors?

Reviewer #2 (Remarks to the Author):

The manuscript by Wahlesdtedt et al builds upon their previous work (Wahlestedt et al Blood, 2013), here the authors convincingly demonstrate that changes associated with HSC aging can be erased by reprogramming to pluripotency. They expand on their previous work by generation of iPSCs using "reprogrammable mice" and add to their previous work only on the basis that this new study corroborates functional potential pre- and post-reprogramming on a clonal level. This is a very clever use of the reprogrammable mouse that exploited completely will provide cues to reversal of aging and for completeness of somatic cell reprogramming. In all there are some issues with the paper associated with readability and nomenclature that need to be addressed.

My biggest concern is the clarity of the way the experiments are presented and explained. There needs to be more detail in the text and figures as to which mice, chimeras or the reprogrammable mouse, are being studied.

- Line 104-105: The authors state that regardless of age a similar number of input HSCs contributed to hematopoiesis in Extended Figure 1a, but the data show a statistically significant increase from aged vs young HSCs. What is the basis for this conclusion?
- Throughout the paper, the use of the term "HSC" is ambiguous. Most of the field would agree that an HSC is a cell with extended self-renewal capacity and complete multilineage regenerative potential within the hematopoietic hierarchy. Often times in this manuscript, the term HSC is used interchangeably with the term clone, which is inaccurate. It would be more accurate if the term HSC was dropped or replaced by the term clone in many instances. This is especially the case when the clone is described as "monopotent myeloid-biased HSC." If a cell is monopotent, it cannot be an HSC as it does not possess multilineage differentiation potential, and it certainly cannot be myeloid-biased as the biased definition is used to describe multilineage repopulating cells that show a greater propensity to generate one lineage over another while still making all the lineages. I would suggest the term myeloid-restricted or myeloid clone based on the functional output of the clone in question.
- Line 134-135: The authors claim to generate iPSCs from aged barcoded HSCs, but according to the methods they sort and reprogram cells with a myeloid-progenitor phenotype: Lin⁻/Sca1⁻/cKit⁺ (line 298). This in part explains why the 5 iPSC lines generated in the paper did not have T cell potential, and only 2/5 showed B cell potential. Please provide justification for using this cell phenotype for reprogramming, OR change the wording to state "generating iPSCs from aged barcoded clone-derived myeloid progenitors."
 - Related to the above point, Line 610 (Table 1 legend) should be changed to "Lineage distribution of barcoded clone-derived myeloid progenitors" as the cells sorted here for reprogramming did not contain HSCs.
- Line 153: This should be Figure 1G, not Figure 2G.
- Many of the figures were difficult to interpret. Within the methods section, lines 244-249, there is a description of cell gating strategies for FACS plots depicted in figures 2 and 3. This is out of place. This data should be included within the Figures themselves, and at the very least should be described in the figure legends. This would add clarity to the figures.
- The transplantation setup of barcoded clones was not well described. Were total cultured

cells, regardless of GFP expression, transplanted after transduction? Were GFP+ cells sorted for transplantation? How many transduced cells were transplanted per mouse? This would be helpful for clarifying the experimental set-up, as well as other unclear metrics like "clone size" used in figure 1d and extended data figure 1. Also, the definition of "clone size" found starting on line 289 was confusing and not very well described. Please clarify this definition.

- In Table 1, what are the statistics being displayed? Are the statistics or GFP+CD45.2+CD45.1- cells? If so, please include this information. If not, why not?
- The description of iPS chimera generation and performing BM transplantations from cells derived from these chimeras is difficult to follow in Table 1 and Figure 2. A diagram depicting the process, not unlike Figure 1A, with descriptions of each stage of the experiment as well as the term used to describe each stage of the experiments (iPS donor mouse, 1' chimera, 1' transplant, primary iPS chimera, 2' BM chimeras) would add a great deal of clarity for the reader.

Related to the above, in Figure 2a, top panel, what is the iPS donor mouse, and why doesn't it contain transduced clone-derived (GFP+) CD3e T-cells? This is especially true given that in Table 1 each primary iPS chimera showed T cell lineage contribution.

Reviewer #3 (Remarks to the Author):

In this manuscript, the authors demonstrated an unbiased tracking of cellular lineage during differentiation and reprogramming using a combined strategy of functional evaluation and cellular barcoding. As claimed by authors, this study provides a fair investigation to tackle the question of chronological versus functional aging of hematopoietic stem cells (HSCs). Indeed, the findings from this study are very intriguing and most experiments are well designed and validated. The paper should be published but certain changes need to be made.

(1) The title is not representative of the data reported in the paper.

(2) Provide details of the sequencing analysis to ensure that the barcode reading was feasible, including the coverage of DNA sequencing.

(3) The justification for the initial transplantation of barcoded HSCs is not addressed carefully enough. The efficiency of transplantation may vary based on aging. If only a subpopulation of aged HSCs is competent for transplantation, the starting point of this study could be biased. This is a critical point to be resolved for this paper to be suitable for publication in Nature Communication.

(4) Data from iPS cells from young HSC could be provided as a positive control, as they are able to generate T cells. This would be interesting because in Supplementary Table 1, for example, one could directly compare the percentages. Of course the positive control of young HSC that the authors have already included are also necessary.

(5) The following sentences in the abstract need to be rewritten in a more concise manner: "However, individual HSCs display significant variability in function even in younger subjects and can also be anticipated to age in an asynchronous manner. Thus, to only evaluate the prospects for rejuvenation in chronologically rather than functionally aged HSCs preclude assessment of the impact of a potential "memory" imparted onto cells with age"

(6) Figure 3 does not provide enough molecular evidence that the HSC and iPSC-HSC from aged donors are different, which does not support the title or description in the abstract. This issue is also raised in the below section.

(7) Aging is a complex process as the author stated in line 35-43, even if we focus only on the HSC lineage. Therefore, when the author states "the reversion of aging," the specific feature of aging needs to be carefully indicated throughout the manuscript. In general, this was relatively carefully described in the main text; however, the abstract may mislead the reader regarding the content of the manuscript. As stated in line 45-48, "We thereafter evaluated the heritability of the HSC aging state by generating induced pluripotent stem cell (iPS) lines from aged HSC clones with a pronounced lineage skewing; a hallmark of HSC aging." The research only focused on the lineage skewing and transcriptional signature, and the authors did not see a difference between iPS lines from young and aged HSCs with regard to lineage skewing and transcriptional signature. The authors should be more specific and tone down the conclusion regarding the aspect of lineage skewing and transcriptional signature in the sentences in lines 49-56: "Our results demonstrate that hematopoiesis emerging from such iPS lines was indistinguishable from that seen in young mice." and "Thereby, by demonstrating full reinstatement of young HSC function from functionally and phenotypically defined aged HSCs, we here firmly establish HSC aging as a reversible cellular state." This specificity should be carefully applied to the main text of the manuscript.

(8) With the same result, the author could present the data in a positive manner, as iPSC reprogramming overcoming the lineage skewing and transcriptional signature of aging. However, the logical structure of the manuscript presents the data as negative, i.e., demonstrating no detectable difference in the resulting iPSC from young and aged HSCs. A precise description of the experimental support for the author's negatively framed conclusion is needed. If the authors state their findings too strongly (e.g., there is no aging-specific memory in any aspect), it immediately begs the following questions: (a) What is the effect of genetic background? (b) What is the aging speed of the T cells derived from iPS cells from aged HSCs? (c) Did they use various ages of donors? (18-24 months in mouse is equivalent to 56-69 years in humans. The authors may want to include in the study cells from older mice.) (d) What are the various epigenetic signatures to support the paper's stated title? and (e) What is the quality of the resulting iPSC? (The researchers make a very strong statement with the paper's title, with two major approaches being used. Their focus is only on T cells. What about the rest of the cells? What were their quality and pluripotency?)

Response to reviewers

Reviewers' comments:

Reviewer #1 (Remarks to the Author):

In this elegant study performed by the Bryder group, the authors have determined on a clonal level that function potential of aged primitive hematopoietic cells can be reset through reprogramming. Using bar-coding techniques to identify lineage output as well as overall contribution, the authors were able to demonstrate that multi-lineage potential, can be restored to cells derived from aged, lineage biased HSCs. The implications of this study are profound as it establishes that cellular reprogramming — which is orchestrated at the level of the epigenome — can reset the epigenome of aged murine HSCs to restore functional potential otherwise lost through aging. In so doing, the study firmly establishes that epigenetic remodeling is a central driver of HSC aging that is amenable to reprogramming.

RESPONSE: We appreciate that the reviewer considers our work both elegant and with important implications.

Comments:

The restoration of overall reconstitution potential is not well characterized in this study. To demonstrate improved reconstitution potential, the authors should perform competitive transplants with the iPS derived HSCs as aged HSCs- with no competition can fully reconstitute an irradiated recipient mouse. The authors should define if transplanted BM from the blastocyst/morula was done in a competitive setting? If so, they should provide the total reconstitution frequency in table 1 for comparison to young transplanted. Also, the data for the control young transplants should be included in table 1.

RESPONSE: The experiments conducted using transplantation of iPS derived cells are by nature competitive. This is because the blastocyst/morula chimeras used for bone marrow transplantation is composed of both endogenous (young) and iPS-derived HSCs. We have in our revised manuscript added % chimerism (relative to primary chimeras) in transplanted hosts (Table 1, rows 636-637) and also clearly stated this in the manuscript text (rows 156-158). In this way, we now correlate to that seen in primary chimeras to directly visualize that the iPS derived hematopoiesis is upheld upon transplantation (which is not to be expected from chronologically aged HSCs).

The authors have previously published on lineage-biased HSCs, defining these populations using CD150/ SLAMF1 expression defining these subsets. The aged population of myeloid biased subsets express high levels of CD150. Were the aged M/E clones expressing high levels of CD150, and during the reprogramming did the expression levels of CD150 change along with lineage contribution?

RESPONSE: As we generated our iPS cells from the progeny of HSCs and not HSCs directly (see also below for more clarification on this issue), any information we may provide would be mainly correlative/indirect. This concern aside, we have in our revised manuscript provided a supplementary figure (Extended Data Figure 2, rows 701-712) containing FACS plots showing the CD48 and CD150 expression among Lin⁻c-Kit⁺Sca-1⁺ cells in the young and aged mice used for barcoding, as well as for the endogenous and iPS derived Lin⁻c-Kit⁺Sca-1⁺ cells in each of the primary chimeric mice. We have also added a section describing this analysis to the manuscript text (rows 173-179). This illustrates not only that the composition of the HSC compartment with regards to CD150 expression is drastically different between the young and aged scenario, but also that the iPS derived Lin⁻c-Kit⁺Sca-1⁺ compartment expresses CD150 to levels comparable to the endogenous young control cells. This

could be taken as evidence that the aged iPS derived cells can establish a primitive hematopoietic compartment that resembles that observed in young mice.

Could the authors clarify the number of barcoded-HSCs transplanted /or starting number of HSCs transduced if the GFP+ cells were not purified before transplant. This point is relevant as the authors state similar numbers of HSCs contribute to overall hematopoiesis in both the young and aged mice.

RESPONSE: We have in our revised manuscript clarified these issues in the methods section (rows 264-274). See also below responses to the first query raised by reviewer #2.

Perhaps a mistake in the methods, but the manuscript states that aged-HSCs were reprogrammed to iPS (line 135) yet in the methods, myeloid progenitors (LSK cKit+Sca1-) are used for the generation of iPS lines. While myeloid progenitors are of course derived from stem cells, the authors should nonetheless clarify. Similarly, in the generation of iPS lines, were there many unique barcodes that were found in the progenitor population that were not read out in the peripheral blood?- i.e were there HSC clones that were active in contributing to primitive compartments, but did not contribute to peripheral blood / erythroid progenitors?

RESPONSE: We thank the reviewer for this observant point. We have attempted to generate iPS cells from both HSCs directly as well as from myeloid progenitors both in this and other (Wahlestedt et al., BLOOD, 2013) work. However, we have never successfully been able to generate iPS cells from stringently purified HSCs, thus the iPS cells here derived from a mixed population of progenitor cells (=the progeny of HSCs). We have in our revised manuscript rewritten this section to clarify the somatic donor cell types used and the reasons underlying the choice of these (lines 134-140). During the course of our work, Guo et al (Cell, 2013) have in association to this point highlighted that iPS capacity from hematopoietic stem/progenitor cells is associated with a privileged state that is connected to the inherent cell cycle properties of the parental cells. The highly dormant state of highly purified HSCs therefore would provide a most plausible explanation for these data, and we have added a reference to this study. We have not directly assessed whether barcodes could be retrieved from other primitive compartments (apart from erythroid progenitors). On the other hand, we feel this information is readily available in previously published work (for instance Lu et al, Nature Biotechnology 2011). We have in our revised manuscript added the total number of generated iPS clones, as well as the number of iPS clones not present in the sequenced lineages to the manuscript text (rows 145-146).

Reviewer #2 (Remarks to the Author):

The manuscript by Wahlestedt et al builds upon their previous work (Wahlestedt et al Blood, 2013), here the authors convincingly demonstrate that changes associated with HSC aging can be erased by reprogramming to pluripotency. They expand on their previous work by generation of iPSCs using “reprogrammable mice” and add to their previous work only on the basis that this new study corroborates functional potential pre- and post- reprogramming on a clonal level. This is a very clever use of the reprogrammable mouse that exploited completely will provide cues to reversal of aging and for completeness of somatic cell reprogramming. In all there are some issues with the paper associated with readability and nomenclature that need to be addressed.

RESPONSE: We appreciate that the reviewer considers our work convincing and that the strategy chosen is clever to approach the question at hand.

My biggest concern is the clarity of the way the experiments are presented and explained. There needs to be more detail in the text and figures as to which mice, chimeras or the reprogrammable mouse, are being studied.

RESPONSE: We are sorry for not being more clear in our original manuscript. We have in connection to the resubmission gone over the manuscripts text and figures and changed the naming conventions for the chimera types to be homogenous throughout the paper. The initial transplant of barcoded HSCs is now called "1° Transfer", the primary iPS chimeras are called "1° iPS chimera" and the chimeras obtained from bone marrow transplantation of the "1° iPS chimera" are labeled "2° BM chimeras". We hope that this will avoid confusion.

- Line 104-105: The authors state that regardless of age a similar number of input HSCs contributed to hematopoiesis in Extended Figure 1a, but the data show a statistically significant increase from aged vs young HSCs. What is the basis for this conclusion?

RESPONSE: The basis for our conclusion that the contribution was "similar" was looking at effect rather than mere statistics, but the reviewer is absolutely correct that slightly more contribution (1.5-fold) was observed from aged HSCs in these experiments, which also reached significance with the statistical test used. We have in our revised manuscript clarified this in the manuscript text (rows 101-104). Still, to us, the most important piece of information here is that the number of contributing clones from aged candidate HSCs is not substantially lower than for young HSCs – an insight which can only be reached using the barcoding technology and which previous studies to a large extent have failed to visualize due to the poor transplantation behavior of aged HSCs. We note however that in one previous study, Verovskaya et al (BLOOD 2013) also failed to reveal a decrease in contributing clones from aged HSCs using oncoretroviral barcoding), which we believe further strengthens our conclusions.

- Throughout the paper, the use of the term "HSC" is ambiguous. Most of the field would agree that an HSC is a cell with extended self-renewal capacity and complete multilineage regenerative potential within the hematopoietic hierarchy. Often times in this manuscript, the term HSC is used interchangeably with the term clone, which is inaccurate. It would be more accurate if the term HSC was dropped or replaced by the term clone in many instances. This is especially the case when the clone is described as "monopotent myeloid-biased HSC." If a cell is monopotent, it cannot be an HSC as it does not possess multilineage differentiation potential, and it certainly cannot be myeloid-biased as the biased definition is used to described multilineage repopulating cells that show a greater propensity to generate one lineage over another while still making all the lineages. I would suggest the term myeloid-restricted or myeloid clone based on the functional output of the clone in question.

RESPONSE: While we of course agree with the reviewer that many of these points are central in HSC biology, we have tried to make our paper readable for a general and not only to a hematopoietic-specific audience, without losing (too much) accuracy. As for the specific phrasing referred to ("monopotent myeloid-biased HSC") – it can be discussed whether the definition of HSCs always would require multipotency and the extent of this multipotency. In this specific instance, we have changed to "myeloid-restricted" – well aware that this might have its' own problems (see below). However, an obvious example would be that of fetal versus adult HSCs, where fetal HSCs carry some potentials that adult HSCs do not. Would we then say that adult HSCs are not HSCs because they cannot generate all hematopoietic subsets? In an extension, if such losses in lineage potential (also for other lineages) are gradual, where would one define the "breakpoint" where cells would no longer be classified as HSCs? A definition is however needed and we feel the one we entertain (self-renewal potential together with continuous/long-term hematopoietic output) is not very controversial. In addition, we feel that the term "biased" is also not that straightforward as proposed, since this would require an ability to measure all output at all points in time and hence would put a lot of emphasis on negative data. Based on the concerns raised here with our nomenclature, we have however gone over the manuscript carefully to try to resolve any obvious ambiguities, while still attempting to preserve

readability.

- Line 134-135: The authors claim to generate iPSCs from aged barcoded HSCs, but according to the methods they sort and reprogram cells with a myeloid-progenitor phenotype: Lin-/Sca1-/cKit+ (line 298). This in part explains why the 5 iPSC lines generated in the paper did not have T cell potential, and only 2/5 showed B cell potential. Please provide justification for using this cell phenotype for reprogramming, OR change the wording to state "generating iPSCs from aged barcoded clone-derived myeloid progenitors."

RESPONSE: Please see above response to reviewer #1 for a description on how iPSC cells were generated and why they were generated from the specific parental cells, as well as how we have clarified this point in our revised manuscript. A first smaller note is that it is not correct that all cells with a Lin-/Sca1-/cKit+ phenotype are myeloid-restricted. Importantly, however, we generated iPSC lines from HSC-transplanted animals long-term after transplantation. Myeloid progenitors by definition do not possess extensive self-renewal potential, hence they have been continuously generated from the cells that were originally barcoded (phenotypic HSCs). The argument would have been valid if we had generated iPSC cells straight from a myeloid progenitor cell, in which case there would have been no way to connect the founder cells to other lineages, as well as in time.

- Related to the above point, Line 610 (Table 1 legend) should be changed to "Lineage distribution of barcoded clone-derived myeloid progenitors" as the cells sorted here for reprogramming did not contain HSCs.

RESPONSE: We do not feel very strongly about this and have changed this into "Lineage distribution of barcoded clones" in our revised manuscript. We feel this is most appropriate because the barcoding was initially done on phenotypic HSCs. Barcoding is a stable event, thus no new barcodes are generated over time but inherited from upstream progenitors (in this case, candidate HSCs).

- Line 153: This should be Figure 1G, not Figure 2G.

RESPONSE: We thank the reviewer for observing this error, which we have corrected.

- Many of the figures were difficult to interpret. Within the methods section, lines 244-249, there is a description of cell gating strategies for FACS plots depicted in figures 2 and 3. This is out of place. This data should be included within the Figures themselves, and at the very least should be described in the figure legends. This would add clarity to the figures.

RESPONSE: We agree that this would add to an easier understanding of these figures and have moved the gating strategy details from the methods section to the figure legends.

- The transplantation setup of barcoded clones was not well described. Were total cultured cells, regardless of GFP expression, transplanted after transduction? Were GFP+ cells sorted for transplantation? How many transduced cells were transplanted per mouse? This would be helpful for clarifying the experimental set-up, as well as other unclear metrics like "clone size" used in figure 1d and extended data figure 1. Also, the definition of "clone size" found starting on line 289 was confusing and not very well described. Please clarify this definition.

RESPONSE: We have in our revised manuscript clarified these issues in the methods section (rows 264-274 for the initial transduction and transplantation experiments and rows 306-316 for a definition of clone sizes).

- In Table 1, what are the statistics being displayed? Are the statistics for GFP+CD45.2+CD45.1- cells?

If so, please include this information. If not, why not?

RESPONSE: The lineage potential of the HSC clones used for iPS derivation was derived from barcode sequencing of the 1^o transplanted mouse, while the data from primary and secondary iPS chimeras is expressed as frequencies among the CD45.2+CD45.1- fraction (iPS-derived fraction). The reason for omitting GFP from analysis in primary and secondary iPS chimeras was due to a large degree of viral silencing in connection to the iPS derivation. This is however not an issue with the actual barcodes, as they are stably incorporated into the genomes of the host cells and extracted by PCR. We have clarified this in the legend to Table 1 (rows 631-641).

- The description of iPS chimera generation and performing BM transplantations from cells derived from these chimeras is difficult to follow in Table 1 and Figure 2. A diagram depicting the process, not unlike Figure 1A, with descriptions of each stage of the experiment as well as the term used to describe each stage of the experiments (iPS donor mouse, 1' chimera, 1' transplant, primary iPS chimera, 2' BM chimeras) would add a great deal of clarity for the reader.

RESPONSE: In an early draft of our manuscript, we discussed whether we should include such a figure. As it takes up quite a lot of space, we finally decided not to include it. However, we feel based on the comments obtained from all of the reviewers, that incorporating such a description could help clarifying how the experiments were done (rather than confining this information to the methods section, as was done in our original manuscript). We have therefore added a revised figure 1, which shows the flow of our experiments as well as what part(s) of the experimental setup is described and in what figure. Please note that we now have removed panel A from the old Figure 1, since it is now an integrated part of the new figure. Lastly, we have gone over the manuscript text and figures and changed the naming conventions for the chimera types to be homogenous throughout the paper. The initial transplant of barcoded HSCs is now called "1^o Transfer", the primary iPS chimeras are called "1^o iPS chimera" and the chimeras obtained from bone marrow transplantation of the "1^o iPS chimera" are labeled "2^o BM chimeras".

Related to the above, in Figure 2a, top panel, what is the iPS donor mouse, and why doesn't it contain transduced clone-derived (GFP+) CD3e T-cells? This is especially true given that in Table 1 each primary iPS chimera showed T cell lineage contribution.

RESPONSE: As a somatic donor for iPS generation, we used a recipient mouse transplanted with barcoded aged HSCs that completely lacked donor (aged-derived) T cells. Therefore, the re-generation of T cells in the iPS chimeras shown in Table 1 is a main functional correction in our work. We have clarified this further in the manuscript text to make this more explicitly understood (rows 134-135).

Reviewer #3 (Remarks to the Author):

In this manuscript, the authors demonstrated an unbiased tracking of cellular lineage during differentiation and reprogramming using a combined strategy of functional evaluation and cellular barcoding. As claimed by authors, this study provides a fair investigation to tackle the question of chronological versus functional aging of hematopoietic stem cells (HSCs). Indeed, the findings from this study are very intriguing and most experiments are well designed and validated. The paper should be published but certain changes need to be made.

RESPONSE: We appreciate that the reviewer finds our study/experiments well designed and validated.

(1) The title is not representative of the data reported in the paper.

RESPONSE: We chose a title that we feel represents our findings. If the reviewer strongly disagrees, it

would have been helpful to suggest an alternative title. We still feel that our chosen title fits the experiments/results reported in this manuscript.

(2) Provide details of the sequencing analysis to ensure that the barcode reading was feasible, including the coverage of DNA sequencing.

RESPONSE: This has been added to the methods as the new Supplementary Table 3 (referral at rows 301-303).

(3) The justification for the initial transplantation of barcoded HSCs is not addressed carefully enough. The efficiency of transplantation may vary based on aging. If only a subpopulation of aged HSCs is competent for transplantation, the starting point of this study could be biased. This is a critical point to be resolved for this paper to be suitable for publication in Nature Communication.

RESPONSE: We demonstrate that aged HSCs contribute similarly (and even slightly more so) to mature output as young HSCs (in terms of frequency), although they do yield dramatically different types and magnitudes of hematopoietic output. We of course cannot say anything about cells that we cannot read out. Based on the reviewer's comments, we think this important to emphasize, and we have added a sentence on this in our revised manuscript (rows 202-206). However, our experiments do provide information on reversibility, as the design of our work allows us to directly assess this. While it is for sure true that native hematopoiesis and hematopoiesis after transplantation might be fundamentally different, we have in our work used the established paradigm that aged HSCs do not produce the same type of mature cell output as young HSCs following transplantation, and that aged HSCs display distinct transcriptional changes – which we also confirm in the work here.

(4) Data from iPS cells from young HSC could be provided as a positive control, as they are able to generate T cells. This would be interesting because in Supplementary Table 1, for example, one could directly compare the percentages. Of course the positive control of young HSC that the authors have already included are also necessary.

RESPONSE: We have consistently used the blastocyst/morula derived cells as control for young cells, rather than chimeras obtained from iPS cells with a young origin. We feel this is the most stringent control as it will also circumvent any potential individual variation, as well as variations caused by environment. While the use of young iPS cells would have doubled the amount of work, which has already been extensive, we feel that this would have added little to the questions we ask.

(5) The following sentences in the abstract need to be rewritten in a more concise manner:
“However, individual HSCs display significant variability in function even in younger subjects and can also be anticipated to age in an asynchronous manner. Thus, to only evaluate the prospects for rejuvenation in chronologically rather than functionally aged HSCs preclude assessment of the impact of a potential “memory” imparted onto cells with age”

RESPONSE: We struggled quite a lot with this phrasing and agree it is not perfect. Because of the concerns raised by the reviewer we have changed this section in our revised manuscript into “However, individual HSCs display significant variability in function and might also age in an asynchronous manner. Thus, evaluating rejuvenation in chronologically rather than functionally aged HSCs fails to reveal a potential “memory” that may arise as cells age.” (rows 37-41).

(6) Figure 3 does not provide enough molecular evidence that the HSC and iPSC–HSC from aged donors are different, which does not support the title or description in the abstract. This issue is also raised in the below section.

RESPONSE: We demonstrate functional differences between aged barcoded HSCs before and after iPS derivation of the progeny of originally barcoded HSCs. We have attempted to assess molecular features attainable on the very low numbers of relevant cells that can be extracted from these animals. Taken together, the data reveal similarities of aged iPS derived HSCs to young HSCs by all criteria we evaluated. See further descriptions below.

(7) Aging is a complex process as the author stated in line 35-43, even if we focus only on the HSC lineage. Therefore, when the author states “the reversion of aging,” the specific feature of aging needs to be carefully indicated throughout the manuscript. In general, this was relatively carefully described in the main text; however, the abstract may mislead the reader regarding the content of the manuscript. As stated in line 45-48, “We thereafter evaluated the heritability of the HSC aging state by generating induced pluripotent stem cell (iPS) lines from aged HSC clones with a pronounced lineage skewing; a hallmark of HSC aging.” The research only focused on the lineage skewing and transcriptional signature, and the authors did not see a difference between iPS lines from young and aged HSCs with regard to lineage skewing and transcriptional signature. The authors should be more specific and tone down the conclusion regarding the aspect of lineage skewing and transcriptional signature in the sentences in lines 49-56: “Our results demonstrate that hematopoiesis emerging from such iPS lines was indistinguishable from that seen in young mice.” and “Thereby, by demonstrating full reinstatement of young HSC function from functionally and phenotypically defined aged HSCs, we here firmly establish HSC aging as a reversible cellular state.” This specificity should be carefully applied to the main text of the manuscript.

RESPONSE: We fully agree that any scientific paper needs to be balanced and rather try to describe what was done, why, the results and the authors’ interpretations. In our work, we compare the HSCs derived from aged iPS cells to young (endogenous) HSCs, and not to HSCs derived from young iPS cells. We believe that this is clearly stated throughout the manuscript and feel that this is the most stringent control (as one could for instance have imagined that iPS derived hematopoiesis, regardless of parental age, would associate with distinct hematopoietic features). It is true that we do not evaluate all aspects of aging, but rather focus on a few of these. However, we feel this is justified as the features we assess are also measurable properties. As stated in the response to reviewer 2, our work/approach for instance cannot evaluate cells that do not produce output (for instance senescent cells) – and we have in our manuscript added a sentence on this (rows 202-206). At the same time, when sending out our manuscript to obtain feedback from scientists outside of our field, we obtained quite a lot of concerns about the readability when being too detailed. However, we agree that in the specific context of the abstract, the sentence “firmly establish...” should be changed. As such, the new final sentence of our abstract reads “Thereby, our data provide direct support to the view that several key functional attributes of HSC aging can be reversed.” (rows 52-53).

(8) With the same result, the author could present the data in a positive manner, as iPSC reprogramming overcoming the lineage skewing and transcriptional signature of aging. However, the logical structure of the manuscript presents the data as negative, i.e., demonstrating no detectable difference in the resulting iPSC from young and aged HSCs. A precise description of the experimental support for the author’s negatively framed conclusion is needed. If the authors state their findings too strongly (e.g., there is no aging-specific memory in any aspect), it immediately begs the following questions: (a) What is the effect of genetic background? (b) What is the aging speed of the T cells derived from iPS cells from aged HSCs? (c) Did they use various ages of donors? (18-24 months in mouse is equivalent to 56-69 years in humans. The authors may want to include in the study cells from older mice.) (d) What are the various epigenetic signatures to support the paper’s stated title? and (e) What is the quality of the resulting iPSC? (The researchers make a very strong statement with the paper’s title, with two major approaches being used. Their focus is only on T cells. What about the rest of the cells? What were their quality and pluripotency?)

RESPONSE: We agree that we could have used a more “positive” phrasing on several locations in our original manuscript and have therefore in our revised version attempted to rephrase this throughout the text. We compare aged iPS derived hematopoiesis to the hematopoiesis of a fully “unperturbed” system (=the young blastocyst/morula derived compartment, see also above). As for (a), we compared hematopoiesis throughout primary transplants, primary chimeras and secondary chimeras, during which we observe a re-instatement of hematopoietic potential. We have not assessed different genetic backgrounds in the same system – in fact, it was a deliberate choice to approach inbred mouse strains for our work, as larger genetic differences potentially could introduce difficulties in interpretation of data. Maybe more importantly, our studies approach the questions at hand in a kinetic fashion – hence, the genetic component is the same for most key comparisons (primary BM chimeras, primary blastocyst/morula chimeras and secondary BM chimeras). As for (b) – we have as described evaluated T lymphopoiesis in primary chimeras (11-14 weeks of age) and after transplantation (18-40 weeks post transfer), without seeing any evidence of accelerated T cell aging. As for (c), we generated iPS cells from candidate HSCs obtained from 23 months aged mice, which were barcoded, transplanted, extracted long-term after transplantation, which was followed by iPS derivation. The proliferative stress of the transplantation in combination with the long-term readout in fact might suggest that the cells were substantially “older” than indicated by merely their chronological age. However, most importantly, the cells behave old by the functional criteria evaluated. As for (d), our title “Direct evidence for rejuvenation of somatic stem cell aging” does not necessarily involve epigenetics. We use gene expression as a proxy for gene/genome regulation, thus this is the only thing we can describe/measure. To obtain reliable epigenetic signatures is not attainable on the very low cell numbers that can be recovered. As for (e) – our work focused on hematopoiesis, which is our area of expertise, and we used iPS cells as a tool to approach questions in this system (and not any other organ/tissue system).

Reviewers' comments:

Reviewer #1 (Remarks to the Author):

The authors have done a great job addressing the points we've raised as well as those raised by the other reviewers. as a result the manuscript is much improved and I believe ready for publication.

Reviewer #2 (Remarks to the Author):

The authors have addressed most of my specific concerns that I raised initially, adding some clarity to the manuscript and experimental approaches. However, in the process of their revisions the authors have added additional confusion and sloppiness to their new explanations and the reworked figures. In the end, I don't think their revisions have improved the manuscript, as it is still difficult to read due to lack of clarity. I do believe that the experimental work is of good quality, the conclusions are largely sound, and data is worthy of publication, but the manuscript needs several improvements. Most of these are associated with improving readability, clarity, and consistency in regards to notation and nomenclature.

As an aside, is there a word or character limit here that the authors are trying to stay under? This article seems excessively short, and would benefit greatly from added explanation.

Specific remarks:

Lines 100-103: Here the authors directly address the issue raised regarding the increased representation of aged barcoded clones. Although they addressed the issue accurately, it seems that their point—as indicated by their response to the reviewer—is that they were glad that there were not fewer aged barcoded clones than young clones. If this is true, the authors should just say this instead of maintaining that the input is "similar." In fact, it seems to me that there is no need for statistical analysis here, as the major point is that there are plenty of detectable barcodes from both young and aging input cells. Suggesting that they are similar and performing statistical analysis to demonstrate that they are different is counter productive. I suggest getting rid of the statistics, as it adds no value to the data, or support to the author's point. Perhaps it would be helpful if they cited Verovskaya et al (2013) next to this point in the text as they did in their responses to reviewers.

Lines 139-152: The authors now include information about how many barcoded iPS lines they generated. They explicitly state they generated 20 iPS lines, 5 of which were used for study in the rest of the manuscript, 8 of which were "not represented in prior sequencing," and 7 others which were unmentioned. Could they please clarify what "not represented in prior sequencing" means? And what are the other 7 lines that were not mentioned at all? If the authors feel the need to bring this information up, then they should follow up on it's explanation, even if that explanation is only "we only used the following 5 lines for further

experiments specifically because they lacked T cell potential after 1' transplant."

Lines 149-152: The authors state that iPS clone-derived myeloid, B, and T cells frequencies were similar to endogenous HSC mature cell output, but this comparison is not shown in Table 1. The authors should consider adding this data if they plan to discuss it.

Line 160: The authors state here "young control HSCs." Are these supposed to be the endogenous (CD45.2+CD45.1+) HSC? This should be explicitly stated as it is in line 165.

Line 173: The authors state "investigated the frequency of HSCs in chimeras." Which chimeras? The iPS chimera's? The 2' BM chimeras? Please specify explicitly.

Lines 181-184: the authors begin talking about mid-aged HSCs. Where did these mid-aged HSCs come from? Are these data from another study? How old are these mid-aged HSCs? Please explain.

Line 265: The authors include data in their methods section stating 29% and 31% transduction efficiency. Perhaps this data would fit well with figure 1 or Figure 2.

Figure 1: The revised figure 1, while appreciated, is sloppy. 4 of 6 panels in Figure 1 reference back to Figure 1, where as in most cases these Figure 1 references should state Figure 2. Please revise. Additionally, some indication of sorting and transplantation strategy, as listed in the methods section would function nicely as the legend for figure 1. Also, this reviewer doesn't feel that the reclassification strategy of "1' transplant" "1' iPS chimera" and "2' BM transplant" is better than what they had initially. This nomenclature is not maintained at all throughout the main manuscript or the comments to reviewers (compare figure 1, with figure 3 legend, with response to reviewer). The authors need to take greater care in being consistent with these labels, especially given the relative complexity of their experimental setup. This reviewer suggests the terms "1' transplant" "iPS chimera" and "2' transplant."

In the responses to reviewer section, the authors suggest their own definition for HSCs as a cell with "self-renewal potential together with continuous/long-term hematopoietic output." This definition is perfectly acceptable, but it would behoove the authors to include this definition in the actual text so that the readers, and not just reviewers, understand the authors' definition of HSCs.

Somewhere in either Figure 3 or Table 1, could the authors demonstrate the overall iPS-derived contribution compared to endogenous cell contribution to the peripheral blood/thymic t cells (ie a chart showing CD45.1-CD45.2+ cells vs CD45.1+CD45.2+ cells)?

Figure 4b y-axis: HSC frequency of what? Lin-c-Kit+ cells? Total BM? Please define more clearly.

Table 1: What is the relative chimerism following BMT column? What are the units for the values found in this column? Are they percents? I don't believe they could be because a %

chimerism could never exceed 100%. Please clarify.

Reviewer #3 (Remarks to the Author):

The manuscript has been carefully revised. Details of the sequencing analyses have been included and the transplantation setup of barcoded clones has been fairly well described. However, the revised manuscript raised the following minor concerns.

1) The title still appears to be too bold of a statement, considering that the main focus is only on T-cells and not other cell lineages. In addition, aging is a complex mechanism with many different components. Here, the experimental focus with regard to aging is on lineage skewing in the hematopoietic system, which is only one component of the aging mechanism. An explanation of the concept of in vitro reprogramming (epigenetic reprogramming during iPSC generation) was also missing, which may lead to a misinterpretation of findings as a spontaneous rejuvenation of somatic cells in vivo (at least, this concept should be in the title).

2) ESC are theoretically highly similar to iPSC cells from young HSC with minimum variability as a positive control, but iPSC from young HSC is the most relevant control to use, if the main goal of the manuscript is to evaluate the reversal of the aging effects of HSC during epigenetic cellular reprogramming (iPSC cell generation from somatic cells). At least a few key data could be shown on iPSC cells from young HSC, or the authors could simply include a better reference from other published papers showing the absence of lineage skewing (with minimal variability) with iPSC cells from young HSC in a similar experimental setting.

Reviewers' comments:

Reviewer #1 (Remarks to the Author): The authors have done a great job addressing the points we've raised as well as those raised by the other reviewers. as a result the manuscript is much improved and I believe ready for publication.

RESPONSE: We thank the reviewer for acknowledging the responses made to his/her previous queries.

Reviewer #2 (Remarks to the Author): The authors have addressed most of my specific concerns that I raised initially, adding some clarity to the manuscript and experimental approaches. However, in the process of their revisions the authors have added additional confusion and sloppiness to their new explanations and the reworked figures. In the end, I don't think their revisions have improved the manuscript, as it is still difficult to read due to lack of clarity. I do believe that the experimental work is of good quality, the conclusions are largely sound, and data is worthy of publication, but the manuscript needs several improvements. Most of these are associated with improving readability, clarity, and consistency in regards to notation and nomenclature. As an aside, is there a word or character limit here that the authors are trying to stay under? This article seems excessively short, and would benefit greatly from added explanation.

RESPONSE: We have taken to us the reviewer's critique and have spent considerable effort going over the manuscript to enhance clarity.

Specific remarks: Lines 100-103: Here the authors directly address the issue raised regarding the increased representation of aged barcoded clones. Although they addressed the issue accurately, it seems that their point—as indicated by their response to the reviewer—is that they were glad that there were not fewer aged barcoded clones than young clones. If this is true, the authors should just say this instead of maintaining that the input is “similar.” In fact, it seems to me that there is no need for statistical analysis here, as the major point is that there are plenty of detectable barcodes from both young and aging input cells. Suggesting that they are similar and performing statistical analysis to demonstrate that they are different is counter productive. I suggest getting rid of the statistics, as it adds no value to the data, or support to the author's point. Perhaps it would be helpful if they cited Verovskaya et al (2013) next to this point in the text as they did in their responses to reviewers.

RESPONSE: The reviewer has interpreted our conclusions correctly on this issue. We have in our revised manuscript simplified the phrasing by saying that both young and aged HSCs contibuted actively to hematopoiesis (lines 106-108), along with a reference to the Verovskaya study. We thank the reviewer for this point, which we feel has made the paragraph much easier to read, while at the same time not emphasizing something we do not want to make a (too) strong statement about.

Lines 139-152: The authors now include information about how many barcoded iPS lines they generated. They explicitly state they generated 20 iPS lines, 5 of which were used for study in the rest of the manuscript, 8 of which were “not represented in prior sequencing,”

and 7 others which were unmentioned. Could they please clarify what “not represented in prior sequencing” means? And what are the other 7 lines that were not mentioned at all? If the authors feel the need to bring this information up, then they should follow up on it’s explanation, even if that explanation is only “we only used the following 5 lines for further experiments specifically because they lacked T cell potential after 1’ transplant.”

RESPONSE: First, we just want to clarify that we did not only characterize lines devoid only in T cell output (M/B/E clones 1-2), but also of three lines devoid of both T and B cells (M/E clones 1-3). However, we agree that the phrasing was a bit unclear in our previous manuscript version, which we feel is maybe connected to the relative complexity of the experiments. We have therefore added detailed information on all barcodes retrieved from the 20 iPS lines generated (lines 346-356).

Lines 149-152: The authors state that iPS clone-derived myeloid, B, and T cells frequencies were similar to endogenous HSC mature cell output, but this comparison is not shown in Table 1. The authors should consider adding this data if they plan to discuss it.

RESPONSE: We have extended Table 1 in our revised manuscript to include detailed information on both iPS-derived and blastocyst/morula-derived (young) cells.

Line 160: The authors state here “young control HSCs.” Are these supposed to be the endogenous (CD45.2+CD45.1+) HSC? This should be explicitly stated as it is in line 165.

RESPONSE: This has now been explicitly clarified.

Line 173: The authors state “investigated the frequency of HSCs in chimeras.” Which chimeras? The iPS chimera’s? The 2’ BM chimeras? Please specify explicitly.

RESPONSE: This has now been explicitly clarified.

Lines 181-184: the authors begin talking about mid-aged HSCs. Where did these mid-aged HSCs come from? Are these data from another study? How old are these mid-aged HSCs? Please explain.

RESPONSE: We thank the reviewer for this comment, that we upon reading our previous version agree was described without little context. We have therefore added a brief sentence (lines 191-193) explaining details on this issue.

Line 265: The authors include data in their methods section stating 29% and 31% transduction efficiency. Perhaps this data would fit well with figure 1 or Figure 2.

RESPONSE: We feel that this would add little to our manuscript and is mainly important for technical reasons. Hence, we favor to keep this as is, without adding the information to any of the figures (which we have struggled quite a lot with to keep from being overcrowded).

Figure 1: The revised figure 1, while appreciated, is sloppy. 4 of 6 panels in Figure 1 reference back to Figure 1, where as in most cases these Figure 1 references should state

Figure 2. Please revise. Additionally, some indication of sorting and transplantation strategy, as listed in the methods section would function nicely as the legend for figure 1. Also, this reviewer doesn't feel that the reclassification strategy of "1' transplant" "1' iPS chimera" and "2' BM transplant" is better than what they had initially. This nomenclature is not maintained at all throughout the main manuscript or the comments to reviewers (compare figure 1, with figure 3 legend, with response to reviewer). The authors need to take greater care in being consistent with these labels, especially given the relative complexity of their experimental setup. This reviewer suggests the terms "1' transplant" "iPS chimera" and "2' transplant."

RESPONSE: We appreciate that the reviewer finds this new figure of relevance to our manuscript. We have in our revised manuscript carefully gone over the text multiple times to try to make sure that the referrals are correct. We also appreciate the reviewer's suggestion on nomenclature, which we feel was highly useful. As a result, we have in our revised manuscript used "1' transplants" "iPS chimeras" and "2' transplants" throughout, and feel this has helped with clarity.

In the responses to reviewer section, the authors suggest their own definition for HSCs as a cell with "self-renewal potential together with continuous/long-term hematopoietic output." This definition is perfectly acceptable, but it would behoove the authors to include this definition in the actual text so that the readers, and not just reviewers, understand the authors' definition of HSCs.

RESPONSE: Whether we can be accredited to have come up with this definition is probably not entirely correct, as it has been rather central (although perhaps not explicitly stated) in previous work on HSC aging. However, we have in our revised manuscript now added a paragraph on it (lines 66-72).

Somewhere in either Figure 3 or Table 1, could the authors demonstrate the overall iPS-derived contribution compared to endogenous cell contribution to the peripheral blood/thymic t cells (ie a chart showing CD45.1-CD45.2+ cells vs CD45.1+CD45.2+ cells)?

RESPONSE: We have now in our revised Table 1 added information on the overall hematopoietic iPS chimerism.

Figure 4b y-axis: HSC frequency of what? Lin-c-Kit+ cells? Total BM? Please define more clearly.

RESPONSE: We have in our revised manuscript (to Figure 4b) added y-axis information to make it clear that the frequency is out of Lin-kit+ cells.

Table 1: What is the relative chimerism following BMT column? What are the units for the values found in this column? Are they percents? I don't believe they could be because a % chimerism could never exceed 100%. Please clarify.

RESPONSE: We have in our revised Table 1 clarified this by also adding side-by-side the absolute frequency of iPS chimerism. We hope that we by this action have clarified that

100% is the levels of chimerism in iPS chimeras, and that values for 2' transplants are relative to this value.

Reviewer #3 (Remarks to the Author): The manuscript has been carefully revised. Details of the sequencing analyses have been included and the transplantation setup of barcoded clones has been fairly well described. However, the revised manuscript raised the following minor concerns. 1) The title still appears to be too bold of a statement, considering that the main focus is only on T-cells and not other cell lineages. In addition, aging is a complex mechanism with many different components. Here, the experimental focus with regard to aging is on lineage skewing in the hematopoietic system, which is only one component of the aging mechanism. An explanation of the concept of in vitro reprogramming (epigenetic reprogramming during iPSC generation) was also missing, which may lead to a misinterpretation of findings as a spontaneous rejuvenation of somatic cells in vivo (at least, this concept should be in the title).

RESPONSE: We thank the reviewer for finding our manuscript carefully revised. As for the title, we agree one should try not to be misleading, which was never our intention. Rather, we have been trying to come up with a title that describes the contextual aspect/s of the work in a broader sense. However, we do agree that we are not approaching all aspects of HSC aging (as we also discuss). We have therefore come up with an alternative title ("**Clonal reversal of aging-associated stem cell lineage bias via a pluripotent intermediate**"). We hope the reviewer can find this title acceptable.

2) ESC are theoretically highly similar to iPS cells from young HSC with minimum variability as a positive control, but iPSC from young HSC is the most relevant control to use, if the main goal of the manuscript is to evaluate the reversal of the aging effects of HSC during epigenetic cellular reprogramming (iPS cell generation from somatic cells). At least a few key data could be shown on iPS cells from young HSC, or the authors could simply include a better reference from other published papers showing the absence of lineage skewing (with minimal variability) with iPS cells from young HSC in a similar experimental setting.

RESPONSE: There is not very much information available on this issue from similar experimental settings. However, we approached the question in previous work with the characterization of 2 iPS lines derived from young blood cells (Wahlestedt et al., *BLOOD* 2013). We have in our revised manuscript added this reference and the reasoning behind our experimental design/comparisons (lines 155-158).